# Analysis of the Energy Loss and Performance Characteristics in a Centrifugal Pump Based on Sinusoidal Tubercle Volute Tongue

**DOI:** 10.3390/e25030545

**Published:** 2023-03-22

**Authors:** Peifeng Lin, Chunhe Wang, Pengfei Song, Xiaojun Li

**Affiliations:** Key Laboratory of Fluid Transmission Technology of Zhejiang Province, Zhejiang Sci−Tech University, Hangzhou 310018, China

**Keywords:** bionics, enstrophy, centrifugal pump, volute tongue

## Abstract

The energy loss inside a centrifugal pump has a significant effect on its performance characteristics. Based on the structural characteristics of the humpback pectoral fin, a new tongue was designed to improve the performance of the centrifugal pump. The influence of three sinusoidal tubercle volute tongues (STVT) and one original volute tongue (OVT) on energy dissipation using the enstrophy analysis method was investigated. To accomplish this, the pressure fluctuations and performances of four centrifugal pumps were analyzed. The results indicate that enstrophy is primarily distributed at the impeller outlet and near the tongue. The total enstrophy of the profiles of STVT was smaller than that of the profiles of OVT. This difference was more obvious near the tongue. The reductions in the total enstrophy of the pumps were 8% (STVT−1), 8.2% (STVT−2), and 9% (STVT−3). The pressure fluctuations of the STVT profiles also decreased to different degrees. The average pressure fluctuations at the monitoring points decreased by 20.6% (STVT−1), 21.7% (STVT−2), and 23.3% (STVT−3). The performances of the bionic retrofit pumps increased by 1.5% (STVT−1), 2% (STVT−2), and 2.45% (STVT−3) under the design flow rate. This study guides the structural optimization of pumps.

## 1. Introduction

A centrifugal pump is a general−purpose energy conversion machine that is widely used in various sectors, including the energy, chemical machinery, aerospace, and agriculture industries, and particularly in national defense equipment. Pump performance is closely related to improvements in production efficiency. Thus, the development of high−efficiency and high−performance centrifugal pumps has been the main goal of the pump industry and has important significance for national energy utilization and economic development.

Various scholars have made significant progress in the research and analysis of the internal energy loss characteristics of centrifugal pumps. The energy loss of a centrifugal pump has an impact on its performance. Further understanding of the energy loss mechanism can guide the structural design and optimization of centrifugal pumps [1]. Wu et al. [2] modified the trailing edge of blades on the suction plane of mixed−flow pumps and introduced a local Eulerian head to reveal the energy−loss pattern along the flow direction of the blades inside the impeller. Hou et al. [3] focused on the magnitude and location of the irreversible hydraulic loss of a two−stage liquefied natural gas (LNG) cryogenic submersible pump based on the entropy production theory. Singh et al. [4] investigated the effect of impeller rounding using a mixed−flow pump as a research model based on the premise of hydraulic performance optimization. Kara Omar et al. [5] combined theoretical and empirical formulations of the energy−loss equation to write a program for studying the performance of a centrifugal pump. Enstrophy can effectively analyze the motion of turbulent fluids and the loss of energy [6]. Lin et al. [7] used the enstrophy dissipation method to investigate the energy loss mechanism of pumps as turbines (PAT) under different flow conditions. They found that this method could accurately predict the location of the hydraulic loss.

The field of bionics has widened and attracted the attention of several experts [8,9,10,11,12,13]. For instance, the sinusoidal tubercles that grow on the pectoral fins of whales allow them to swim at high speeds in seawater. Shi et al. [14] used this feature to design tidal turbine blades with sinusoidal tubercles. The results indicated that the bionic blades promoted the hydrodynamic performance of the hydrofoil and improved the lift−to−drag ratio. Guo et al. [15] found that biologically protruding waveguide blades can effectively improve the humping characteristics of pump turbines. Li et al. [16] found that the bionic sinusoidal tubercle trailing edges of the blade can significantly minimize the energy loss and pressure pulsation of the centrifugal pump and improve the operating performance of pumps.

Inspired by bionics, this sinusoidal shape of the nodule was applied to the volute in this study. Lin et al. [17] designed three shapes of bionic tongues to study the effect of different amplitude sinusoidal tubercle shapes on the performance of the volute tongue. The authors analyze the effects of the rotor−stator interaction and energy loss characteristics to study the reason why the bionic tongue affects the centrifugal pump performance. A feasible solution is provided to enhance the pump’s stability and reduce the vibration noise of the centrifugal pump.

## 2. Main Geometry

### 2.1. Test Model

Herein, the data of the low−specific−speed centrifugal pump adopted by Kelder et al. [18] (as shown in Figure 1) were used to verify the numerical method. The flow rates of the experiments ranged from 50% to 140% of the design flow rate. The static pressure and velocity values were obtained using U−tube manometers and a Laser Doppler Velocimeter (LDV) measurement system. The centrifugal pump consisted of a cylindrical volute tongue with a tongue nose diameter of 2 mm, an impeller composed of seven blades with a thickness of 2 mm, and a constant blade angle of 70°. The design flow rate $Q_{d}$ and angular velocity $V_{\omega }$ are $28.8\;m^{3}/h$ and 4.2 rad/s. The main geometric parameters of the experimental pump are shown in Figure 1 and are listed in Table 1.

### 2.2. Volute Tongue

Figure 2 and Table 2 show the structures and parameters of sinusoidal tubercle volute tongues (STVT) with different amplitudes devised based on the original volute tongue (OVT), where λ is the distance between two adjacent sinusoidal nodules and A is the sinusoid’s amplitude. The bionic volute tongues were designed to contain four tubercles. Because the volute tongue’s width (*C*) is 25 mm, the value λ was 7.5 mm. The values of A in sinusoidal tubercle volute tongues were 1 mm, 2 mm, and 3 mm. The longitudinal length L of the sinusoidal grooves near the tongue was 15 mm. Along the direction of the tongue profile line, the groove depth decreases gradually from 2A to 0.

## 3. Numerical Investigation

### 3.1. Numerical Method

The internal flow of the centrifugal pump can be regarded as a three−dimensional viscous incompressible flow, so the governing equations include an equation for the conservation of mass and an equation for the conservation of momentum.

The equation for conservation of mass:(1)$$\frac{\partial u_{i}}{\partial x_{i}}=0$$

The equation for conservation of momentum:(2)$$\frac{\partial u_{i}}{\partial t}+u_{i}\frac{\partial u_{j}}{\partial x_{j}}=\nu \frac{\partial^{2}u_{i}}{\partial x_{j}\partial x_{j}}-\frac{1}{\rho }\frac{\partial p}{\partial x_{i}}+s_{i}$$
where ν, ρ, and $s_{i}$ represent kinetic viscosity, fluid density, and broad source terms, respectively.

In this study, the DES turbulence model is used to simulate the pump. The DES model can be expressed as:(3)$$\frac{D_{v}}{D_{t}}=c_{b1}s\nu +\frac{1}{\sigma }\left[\nabla \cdot \left(\left(\nu +V\right)\nabla \nu +c_{b2}\left(\nabla \nu \right)\right)\right]-c_{w1}f_{w}{\left(\frac{\nu }{d_{w}}\right)}^{2}$$
where $d_{w}$ represent the characteristic length of the turbulence model.
(4)$$\nu_{t}=\nu f_{v1}$$
(5)$$f_{v1}=\frac{x^{3}}{x^{3}+c_{v1}^{3}}$$
(6)$$x\equiv \frac{\bar{\nu }}{\nu }$$
where ν represent the coefficient of kinematic viscosity.
(7)$$s\equiv f_{v3}s+\frac{\nu }{k^{2}d_{w}^{2}}f_{v2}$$
(8)$$f_{v2}={\left(1+\frac{x}{c_{v2}}\right)}^{-3}$$
(9)$$f_{v3}=\frac{\left(1+xf_{v1}\right)\left(1-f_{v2}\right)}{x}$$
(10)$$f_{w}=g{\left(\frac{1+c_{w3}^{6}}{g^{6}+c_{w3}^{6}}\right)}^{\frac{1}{6}}$$
where s denotes the absolute value of the local eddy volume.

### 3.2. Numerical Solution

ANSYS CFX 18.0 was selected here to simulate a flow in the centrifugal pump. Set the rate of mass flow and static pressure as boundary conditions at the outlet and inlet, respectively. Referring to the results that were presented by Kelder et al. [18]. All are in a stationary computational domain except for the impeller, which is set in a rotating computational domain. The roughness was ignored, which means that all walls were non−sliding and smooth. Two dynamic−static interfaces were established in the computational domain: one between the inlet section and the impeller domain, and another between the impeller and the volute domain. The Euler implicit was chosen as the time derivative, and the second−order upwind was chosen as the advection term. Set one cycle of impeller rotation to 360 steps, i.e., 1° represents a one−time step.

A detached−eddy simulation (DES) model based on the SST k−ω model was applied to simulate the transient flow in the pump. The flow near the wall was determined by using an automatic near−wall treatment method. The outflow pressure and other parameters were monitored to determine whether the flow was fully developed. Up to twelve revolutions of pump rotation were calculated, and the last four were used to obtain the statistical results [19]. The results in this study were then obtained or (temporal and/or spatial) averaged from the transient simulation, i.e., DES. The numerical solution is in agreement with Lin et al. [17], and the detailed numerical solution can be found therein.

### 3.3. Computational Mesh

Figure 3 shows the structured hexahedral mesh of the entire calculation area of the pump, the wall of the impeller blade, and bionic tongues. The average value of $y^{+}$ in the model is 7.2; the average values of $y^{+}$ in the impeller and volute are 7.7 and 6.9, respectively. When the maximum value of $y^{+}$ in the model is 17.4, the maximum values of $y^{+}$ in the impeller and volute are 18.4 and 15.2, respectively. As shown in Table 3, the mesh scheme in Case 3 is optimal and can be used for further transient simulation.

### 3.4. Validation of Calculation Results

As shown in Figure 4, the numerical simulation of the head curve coincided with the actual experimental head curve under small flow conditions, but there was a small difference when $Q/Q_{d}>1.0$.

The pressure distributions of the four pumps are represented by dimensionless pressure values. The static pressure coefficient is expressed using Δp and is defined as follows:(11)$$\Delta p=\frac{p-p_{inlet}}{\rho {\left(V_{\omega }r_{TE}\right)}^{2}}$$
where $r_{TE}$, ρ, and $V_{\omega }$ represent the blade−tip radius, fluid density, and angular velocity of the pump, respectively.

A comparison between the numerically simulated and experimental static pressures at 1.0*Q_d_* flow condition is shown in Figure 5; Figure 5a,b shows that the simulation of static pressure around the impeller and the volute wall agrees well with the experimental. Therefore, the accuracy of the numerical method selection was verified.

Figure 6 shows a comparison between the numerically simulated and experimentally time−averaged circumferential and radial velocity components on the A, D, F, and H traverses (Figure 1) under a 1.0*Q_d_* flow condition and in good agreement. The accuracy of the numerical method is further illustrated.

## 4. Results and Analysis

### 4.1. Performance Analysis of Four Test Pumps

This section introduces the validation of the numerical calculation results of the model and bionic pumps. Additionally, it analyzes the influence of different amplitude sinusoidal tubercle tongues on the pump’s flow field. This is accomplished by using pressure coefficient distribution and pressure pulsation to explain the performance difference between the bionic and prototype pumps.

#### 4.1.1. Head Coefficient and Efficiency for Four Pumps

Figure 7 shows the curves of the performance of the prototype pump and three bionic pumps under different flow rates (0.2*Q_d_*~1.4*Q_d_*). The head coefficient $\psi^{*}$ is defined as
(12)$$\psi^{*}=\frac{gH}{{\left(V_{\omega }d_{2}\right)}^{2}}$$

Overall, the four model pumps’ performance curves followed the same trend. The performance of different amplitude sinusoidal tubercle pumps was superior to that of the prototype pump. Under the design flow rate, the difference in the head coefficients of the four pumps was not significant, and the differences between the three bionic pumps and the prototype pump were 1.4% (STVT−1), 2.3% (STVT−2), and 2.4% (STVT−3). Under high flow rates, the difference widened slightly, particularly for STVT−3, which increased to 7.8%. Under low flow rates, there was little difference in efficiency between the four pumps. At 0.84*Q_d_*, the bionic pump efficiency was significantly higher than that of the prototype pump, and this difference reached a maximum at 1.32*Q_d_*. At 1.0*Q_d_*, the differences in efficiency between the three different amplitude sinusoidal tubercle pumps and the prototype pump were 1.3% (STVT−1), 1.7% (STVT−2), and 2.1% (STVT−3). The relative increases were 7% (STVT−1), 8.1% (STVT−2), and 9.2% (STVT−3) at 1.32*Q_d_*. The comparison revealed that the performances of all the bionic pumps were better than those of the prototype pump, with the model STVT−3 being the best.

#### 4.1.2. Pressure Coefficient Distribution for Four Pumps

The dimensionless pressure fluctuation coefficient is represented by $C_{p}$, which is introduced to process the pressure signal and then analyze the pressure spectra, which is defined as
(13)$$C_{p}=\frac{\left(p_{i}-\bar{p}\right)}{\frac{1}{2}\rho u_{2}^{2}}$$
where $\bar{p}$ denotes the time−averaged pressure. The sample pressure values at each time step were represented by $p_{i}$. The impeller outlet circumferential speed was represented by $u_{2}$.

Figure 8 shows the distributions of the pressure coefficients (upper left) and velocity streamlines (lower right) at the cross sections of the four pumps. Evidently, the pressure fluctuation is highest near the tongue, downstream of the impeller channel, and in the area of the impeller outlet. Simultaneously, it is noticed that the pressure fluctuation in the tongue region is the strongest, indicating that the flow there is extremely unsteady owing to the influence of the rotor−stator interaction. By observing the distribution of pressure coefficients in the tongue region, STVT−1, STVT−2, and STVT−3 were found to have more uniform pressure coefficient distributions and fewer high−pressure regions. This indicates that the bionic tongue can improve the flow inside the centrifugal pump, making the internal flow field structure more stable with less pressure fluctuation. Among them, the pressure fluctuation at the STVT−3 tongue is the lowest, and the interaction between the rotor and the stator has the least effect on it.

#### 4.1.3. Pressure Fluctuation for Four Pumps

As shown in Figure 9, two monitoring points were set near the tongue to record the transient pressure at each time step during the simulation. P1 was set inside the volute facing the impeller, and P2 was set right on the leading edge of the tongue. Then, we quantitatively analyze the strength of the pressure fluctuation in each pump.

Figure 10 and Figure 11 show the pressure fluctuations in the frequency and time domains at P1 and P2, respectively, under the design condition during two impeller revolutions (2T). In Figure 10, the pressure fluctuations at each monitoring point are periodic, with seven peaks and valleys in each period. Compared with the P2, the fluctuations of pressure at the P1 had lower fluctuation amplitudes. Simultaneously, the pressure fluctuation amplitude of the bionic pumps was lower than that of the prototype pump, with a greater reduction at the monitoring point P2. In the tongue region, the radial gap between the impeller and volute wall is small, the trailing vortex downstream of the impeller impacts the volute, and the fluid velocity gradients are large. All of these factors result in large pressure fluctuations, which are effectively attenuated by the bionic structure. The blade passing frequency (BPF) dominates the pressure fluctuation frequency domain, with relatively low amplitudes at the double and triple BPF, as shown in Figure 11. The amplitudes of the bionic pumps indicate a larger reduction at the monitoring point P2, and the amplitudes at the BPF are much lower than those at the OVT. Table 4 shows the reduction in the pressure fluctuation amplitudes at the BPF for the three bionic pumps. At the monitoring point P2, reductions were 60% (STVT−1), 62% (STVT−2), and 67% (STVT−3). In conclusion, the bionic tongue increased the stability of the pressure and improved the flow structure. Among these, STVT−3 showed the optimum effect.

### 4.2. Enstrophy Analysis of Four Test Pumps

#### 4.2.1. Total Enstrophy Analysis

Enstrophy is a new quantity derived from energy. The higher the value of enstrophy, the faster the energy is dissipated. In turbulence, there are various scales and shapes of vortices, which move, stretch, and deform continuously. Stretching of the vortex tube length results in smaller−scale motions. The process of energy transfer from large scales of motion to small scales is known as the “energy cascade.” Large, low−frequency eddies obtain their energy from the mean flow and each other. Small, high−frequency eddies, on the other hand, lose energy through viscous dissipation. Conversely, enstrophy can be used to characterize the average elongation of the vortex tube, and it is proportional to the turbulent energy dissipation rate ε. Therefore, as the vortex tube stretches, the vortex strength increases. However, the increased vortex strength, in turn, leads to enhanced energy dissipation. The enstrophy was defined as the volume integration of the scalar $\frac{1}{2}\rho \omega^{2}$ [20].
(14)$$\Omega (t)=\frac{1}{2}\int \rho \omega^{2}dv$$
where ρ is the fluid density and ω is the flow vorticity. Ω(t)=0 is a necessary and sufficient condition for the flow to be irrotational. By integrating the dissipation function, the enstrophy is associated with energy dissipation. We can further analyze the distribution of total enstrophy in different pumps, reveal the trend of mechanical energy dissipation, and finally construct a quantitative correlation between the internal vortex and external characteristics, such as the impeller head.

Figure 12 shows the values of total enstrophy in the volute domain in one impeller cycle. One data file was extracted every 12°, for a total of 30. It can be found that the total enstrophy curves of the four model pumps have seven peaks and troughs. This was because the model pump had seven blades. When the blade sweeps across the tongue area, the energy dissipation of the flow increases and reaches a peak, which corresponds to the crest position in the figure. Comparing the total enstrophy of the flow in each model pump, it was found that the curves of the bionic pumps were lower than those of the prototype pump, and the STVT−3 profile had the lowest total enstrophy value. Table 5 shows the time−averaged total enstrophy of each model pump in the $\frac{1}{7}$ impeller cycle and the reduction compared with the prototype pump. It can be observed that STVT−3 exhibited the largest reduction, reaching approximately 10%. The reductions in STVT−1 and STVT−3 were slightly lower than those in STVT−3, both of which were approximately 9%. Compared to the original tongue, the sinusoidal structure can reduce the energy dissipation of the centrifugal pump, and STVT−3 demonstrates the most significant effect.

To visually display the enstrophy of the flow in each model pump, the distribution of enstrophy per unit volume (i.e., $\rho \omega^{2}/2$, simply called enstrophy density) in the flow is analyzed $\frac{1}{2}\rho \omega^{2}$. Figure 13 shows the isosurface distribution of the enstrophy density in the four model pumps, and the value of the isosurface was 65,000. It can be observed that in the impeller domain, the enstrophy of the flow is primarily distributed on the blade pressure side and near the leading edge of the blade. In the volute domain, the enstrophy of the flow was mainly distributed near the tongue and impeller outlet. Higher enstrophy was observed near the tongue. Because the clearance between the volute wall and blade was the smallest, the energy dissipation in this area was the largest. There is little difference in the enstrophy distribution of the four model pumps in the impeller, which shows that the bionic tongue cannot significantly reduce energy dissipation in the impeller. Near the tongue, the enstrophy of bionic pumps was much lower than that of the OVT profile. The OVT profile has extremely high−energy dissipation at the downstream tongue. The energy loss of the STVT profiles was concentrated only on the tongue. The enstrophy profiles of STVT−1 and STVT−2 involve several small areas in this region, whereas the STVT−3 profile has almost no enstrophy. This is similar to the conclusions drawn from Figure 12.

The above results show that the bionic tongue has a significant impact on the flow in the tongue area, particularly downstream of the tongue, and has little impact on the impeller domain. As shown in Figure 14, four sections with *y^+^* of 15, 20, 25, and 30 in the downstream region of the tongue were selected to analyze the enstrophy distribution.

As shown in Figure 12, the enstrophy curve of the flow shows seven blade−passing periods when each impeller rotates by one revolution. Among them, the flow enstrophy of the profile of STVT−3 has the maximum decrease. Therefore, the distribution of the STVT−3 and the OVT profiles enstrophy at six evenly distributed times (T1–T6) in approximately one blade passing period was analyzed, and the rotation angle from Tn−1 to Tn is 9°, as shown in Figure 15.

The enstrophy in the flow of the OVT and STVT−3 profiles is concentrated around the tongue, where large shedding vortices exist and a large energy loss is experienced. In Section 1 and Section 2, the high−energy loss area of the OVT profile first shows an increasing trend and then decreases with time, as shown in Figure 15a,b. At time T1, the energy loss was the smallest. After that, the high−energy loss area extended in the downstream direction of the tongue, reached a maximum at time T4, and then decreased in the reverse direction. This is because in the entire period (T1–T6), the blade sweeps through the tongue, and the distance between the blade and tongue is the shortest at time T3. The clearance between the blade and tongue was the smallest, and the interaction between the rotor and the stator has the greatest influence. This leads to the maximum energy loss at this moment. In Section 3 and Section 4, the energy dissipation of the OVT is significantly weakened in the tongue region. At time T1, only a large area of concentrated energy loss was observed. As time passes, the high−energy loss area moves backward as a whole and slowly splits into several small pieces until it disappears. This is because vortices flow in the downstream direction of the tongue near the wall (*y^+^* < 10). Fluctuations occurred at *y^+^* = 15−20. From time T1 to T4, the vortices moved in the vertical direction to the high *y^+^* section. As the distance from the wall becomes larger, the turbulent fluctuation decreases and the energy loss is further reduced, which results in a much smaller energy loss area in Section 3 and Section 4 than in Section 1 and Section 2. In the horizontal direction, as the blade sweeps across the tongue area, the streamwise vortices move downstream to the tongue and then disappear slowly. This results in a downstream movement of the high−energy dissipation zone of Section 3 and Section 4.

Compared to the enstrophy distribution of the OVT profile, the high−energy dissipation region of the STVT−3 profile is reduced to varying degrees in each cross−section. In Section 1, the high−energy dissipation region of the STVT−3 profile extends along the groove of the sinusoidal tongue, occupying a smaller area than the prototype pump. In Section 2, this phenomenon is even more obvious, with the energy loss area of the STVT−3 profile splitting from the entirety of the OVT profile into a slender strip of area. As described above, enstrophy is directly related to the kinetic energy that corresponds to the dissipation effects in the fluid. The sinusoidal structure of the bionic tongue makes the high dissipation area near the downstream and of the tongue much smaller, resulting in lower energy loss. In Section 3, the high−energy loss area of the STVT−3 profile disintegrates into several small pieces. Compared to the OVT profile, the total area occupied by this region was significantly reduced. In particular, at times T3, T4, and T5, there are almost no high−energy loss areas. However, in Section 4, the high energy loss area of the STVT−3 profile is much weaker than that of the OVT profile in all instances, and large high−energy loss areas exist. This indicates that the bionic tongue has a significant impact on reducing pressure fluctuations in the tongue region, which will also reduce energy loss. Furthermore, this improvement has a relatively small effect in Section 1 and Section 2 near the wall and a significant effect in Section 3 and Section 4.

#### 4.2.2. Enstrophy Transport Analysis

The enstrophy transport equation can be derived according to the vortex dynamics equation. Ignoring the expansion and viscosity fluctuations, it takes the following form [20]:(15)$$\frac{1}{2}\frac{\partial \omega_{i}\omega_{i}}{\partial t}+\frac{u_{j}}{2}\frac{\partial \omega_{i}\omega_{i}}{\partial x_{j}}=P_{\omega }+D_{\omega }+T_{\omega }$$
(16)$$P_{\omega }=\omega_{i}\omega_{j}s_{ij}$$
(17)$$s_{ij}=\frac{1}{2}\left(\frac{\partial V_{i}}{\partial x_{j}}+\frac{\partial V_{j}}{\partial x_{i}}\right)$$
(18)$$D_{\omega }=\frac{\nu }{2}\frac{\partial^{2}\omega_{i}\omega_{i}}{\partial x_{j}\partial x_{j}}$$
(19)$$T_{\omega }=-\nu \frac{\partial \omega_{i}}{\partial x_{j}}\frac{\partial \omega_{i}}{\partial x_{j}}$$

The left side of the above formula represents the growth rate of enstrophy along the particle track. The right−hand side $P_{\omega }$ represents the production of enstrophy, where $s_{ij}$ is the tensor of the strain rate. If the deformation of the flow stretches the vortex tube, the entropy increases, which accelerates the energy dissipation and loss of mechanical energy. $D_{\omega }$ is an enstrophy and molecular diffusion mechanism. $T_{\omega }$ is the molecular viscosity dissipation of enstrophy, which always increase dissipation. All three terms are then given as time−averaged variables from the transient simulation results. As indicated above, the STVT−3 profile had the best improvement effect. Therefore, we studied the energy−loss characteristics of the OVT and STVT−3 profiles according to the enstrophy transport equation.

Figure 16 shows the three terms on the right−hand side of the enstrophy transport equation at the intersection between the midplane and outlet of the impeller (illustrated as a red circle in the pump sketch). It can be observed that all the curves have seven peaks, corresponding to the positions of the seven blades of the model pumps. The first peak was the largest, and then it decreased sequentially. In addition, the peak of the production $P_{\omega }$ is significantly higher than that of $D_{\omega }$ and $T_{\omega }$, which shows that the net enstrophy change is larger than 0, which means there is a mechanical energy loss. Throughout the three figures, the significant reduction in amplitude of all three terms shows that the structure of the STVT−3 bionic tongue can stabilize the flow and decrease the mechanical energy loss near the tongue.

Figure 17 shows the three terms at the intersection between the midplane and volute wall (illustrated as a red curve in the pump sketch). In contrast to the seven peaks of each curve in Figure 16, there was only one peak near the tongue, and the fluctuation near the tongue was much greater. Compared with the impeller outlet area, the amplitude of each peak increased significantly. This can be explained by the vortex tube stretching and the vorticity of the wake vortex shedding from the trailing edge of the blade increasing, which increases mechanical energy dissipation. Compared with OVT, STVT−3 had the largest reduction in the peak of the production term near the tongue. The results indicate that the STVT−3 bionic tongue structure can most significantly reduce the energy loss near the tongue.

Figure 18 shows the isosurface distribution of the enstrophy production in the volute domain of the OVT pump (colored by the velocity magnitude), and the value of the isosurface is 65000. It can be observed that the vortex tube stretching is primarily distributed in the impeller outlet and the area near the partition tongue. Among them, the vortex tube stretching in the tongue area was the most obvious. There is a much higher energy loss in this region than in other areas. Therefore, Figure 19 shows the iso−surface distribution of the product terms for both STVT−3 and OVT at different times (T1−T6) near the tongue to study the time−evolution characteristics of the enstrophy production in the two pumps.

From the figure, the stretching of the vortex tube primarily occurs at the impeller outlet and tongue region, which implies a large amount of energy loss. Observing the distribution of the enstrophy production of the OVT at T1–T6 times, it can be noticed that the stretching of the vortex tube primarily starts at the trailing edge of the blade. The vortex is thrown out of the impeller owing to the inertia of the blade rotation and then hits the tongue region, and this cycle occurs repeatedly. This leads to a general elongation of the average length of the vortex tube near the tongue, which in turn generates a large amount of energy loss.

Comparing the production of the two model pumps at various times, it was found that the production term of the profile of the STVT−3 is much shorter than that of the OVT. Additionally, the distribution of production at the impeller outlet is reduced, which leads to a reduction in the total vortex tube stretch in the tongue region. This shows that the bionic tongue can decrease the stretching of the vortex tube, lessen enstrophy, and finally reduce mechanical energy loss.

## 5. Conclusions

The effects of four tongues on the performance of centrifugal pumps were investigated by numerical simulations. The following conclusions can be drawn:

Through the enstrophy analysis of the four models, it was found that the impeller outlet was the area with the highest energy dissipation, and the energy dissipation near the volute tongue was the highest. The enstrophy near the bionic tongue was low, indicating that it can effectively reduce the energy dissipation of the centrifugal pump.

The total enstrophy reduction rates of the bionic pump were 8%, 8.2%, and 9%, respectively, which are less than those of the original model. By comparing the average pressure fluctuation of the four pumps, it was found that the average pressure fluctuation of the pump equipped with STVT was reduced by 20.6% (STVT−1), 21.7% (STVT−2), and 23.3% (STVT−3) near the monitoring point. Under the 1.0*Q_d_* flow condition, the efficiencies of the three pumps with sinusoidal tubercle tongues of different amplitude increased by 1.5%, 2%, and 2.45%, respectively. Therefore, the bionic tongue can significantly improve the performance of the centrifugal pump, with STVT−3 being the most effective. Therefore, compared with reference [17], it is found that when the number of sinusoidal tubercles is fixed, appropriately increasing the amplitude size can not only improve the performance of the pump but also improve the flow in the pump.

In conclusion, the performance enhancement of the centrifugal pump is more pronounced with the larger amplitude of the bionic tongue tubercles. Although this research has provided the pressure fluctuation and energy loss characteristics of centrifugal pumps, there is still some work to be carried out. The next step involves performing an experimental analysis, adopting a 3D−printing method to design a bionic tongue, and conducting a series of experiments to verify the improvement of the flow field in a centrifugal pump using a bionic tongue.

## Figures and Tables

**Figure 1 entropy-25-00545-f001:**
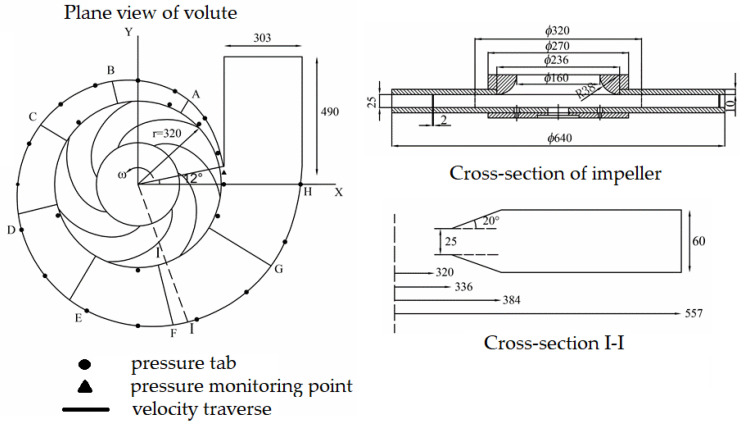
Pump geometry of the centrifugal pump and the locations of velocity and static pressure measurement ([18]. Kelder, J.D.H.; Dijkers, R.J.H.; van Esch, B.P.M.; Kruyt, N.P. Experimental and theoretical study of the flow in the volute of a low specific-speed pump. *Fluid Dyn. Res.*
**2001**, *28*, 267–280. https://doi.org/10.1016/S0169-5983(00)00032-0).

**Figure 2 entropy-25-00545-f002:**
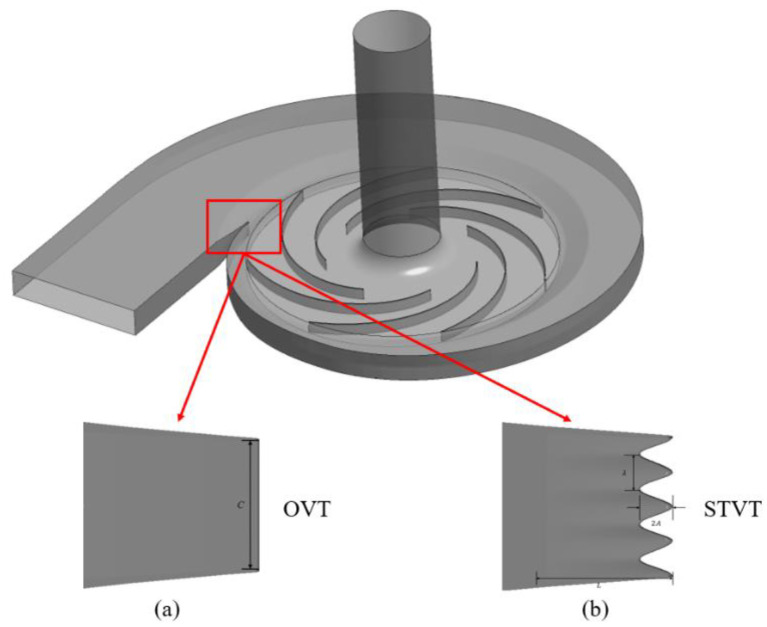
Differences between two types of tongues: (**a**) OVT and (**b**) STVT.

**Figure 3 entropy-25-00545-f003:**
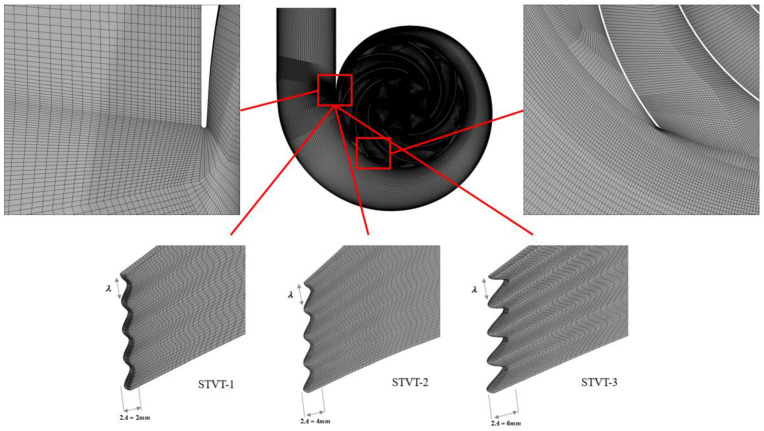
Structure of the hexahedral mesh of the impeller and volute.

**Figure 4 entropy-25-00545-f004:**
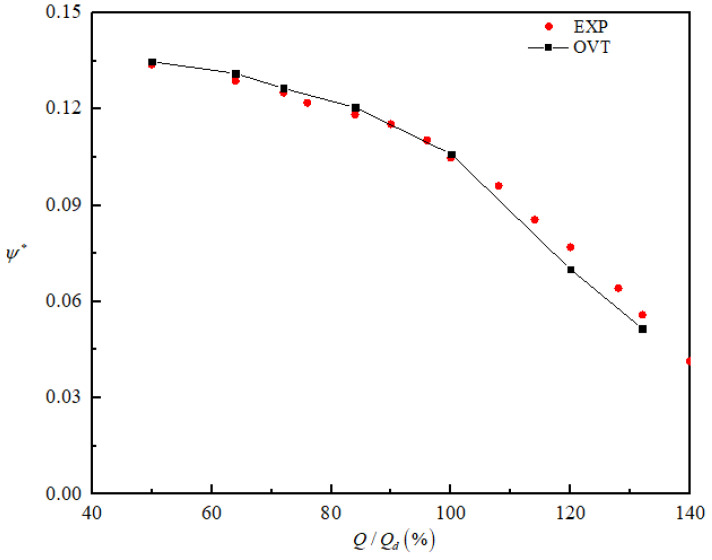
Comparison between numerical and experimental values of the head coefficient under different flow rates.

**Figure 5 entropy-25-00545-f005:**
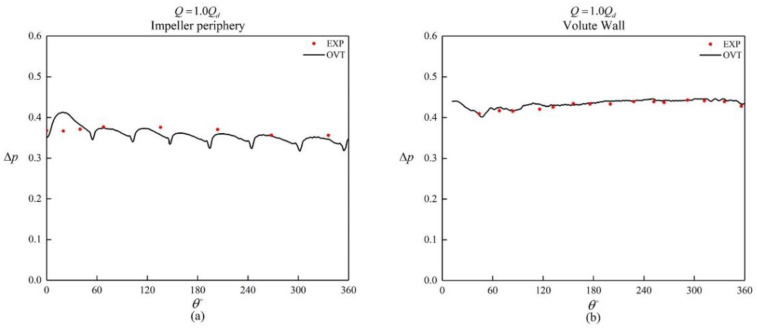
Comparison of predicted and measured non−dimensional static pressure distributions: (**a**) impeller periphery and (**b**) volute wall.

**Figure 6 entropy-25-00545-f006:**
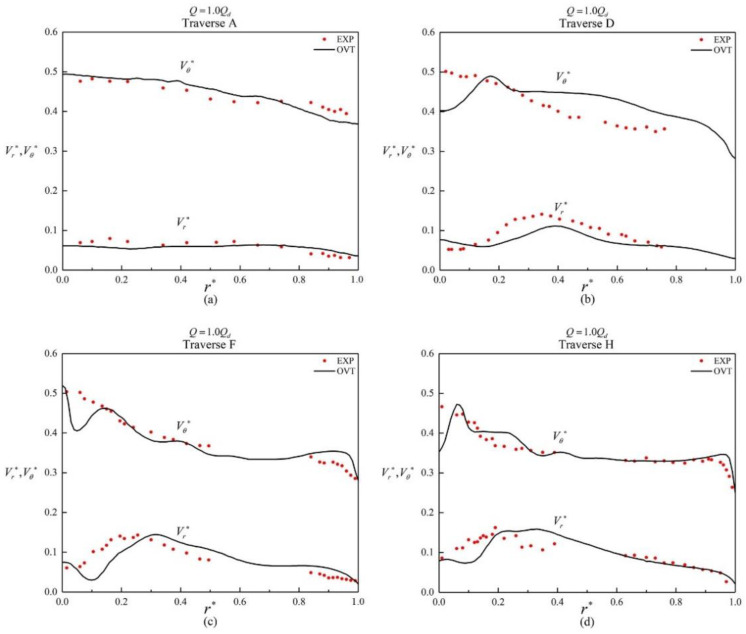
Non−dimensional radial and circumferential velocities in the volute along different traverses at the design flow rate: (**a**) Traverse A; (**b**) Traverse D; (**c**) Traverse F; and (**d**) Traverse H.

**Figure 7 entropy-25-00545-f007:**
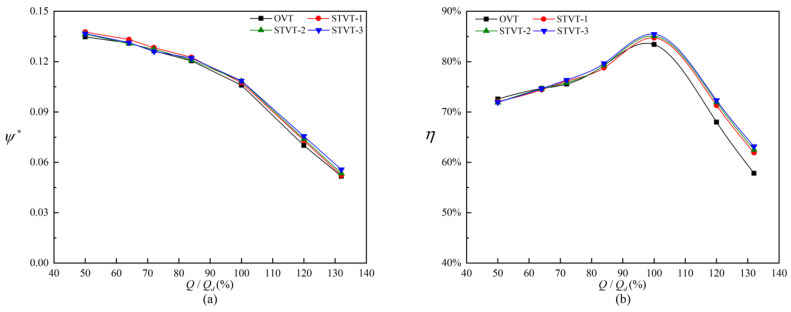
Comparison of predicted performance curves with different tongues: (**a**) head coefficient; (**b**) efficiency.

**Figure 8 entropy-25-00545-f008:**
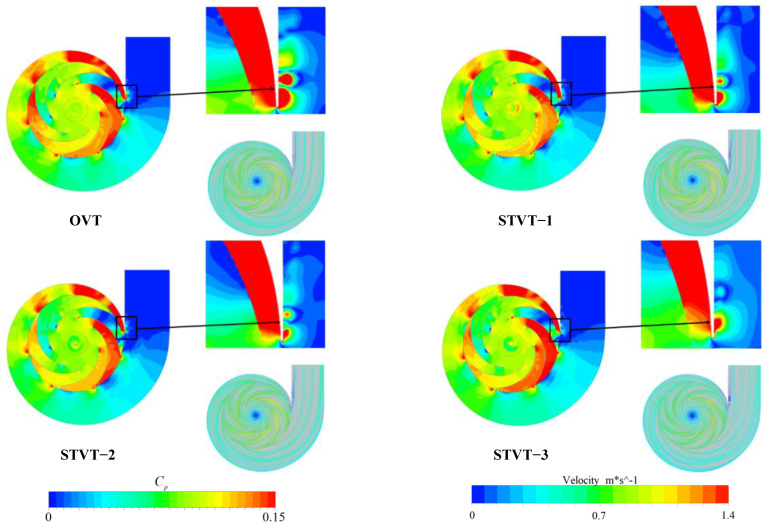
Distribution of pressure coefficients and streamlines at cross−sections in four pumps.

**Figure 9 entropy-25-00545-f009:**
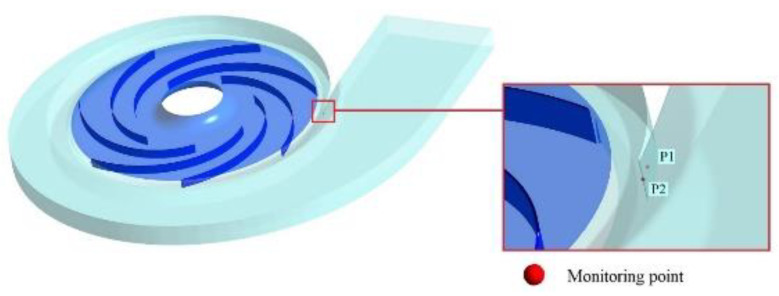
Location of monitoring points.

**Figure 10 entropy-25-00545-f010:**
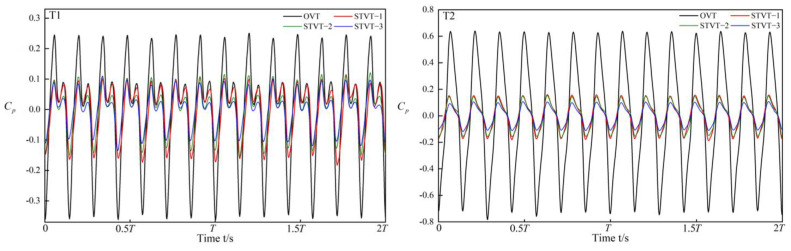
Time−domain pressure fluctuations at points P1 and P2.

**Figure 11 entropy-25-00545-f011:**
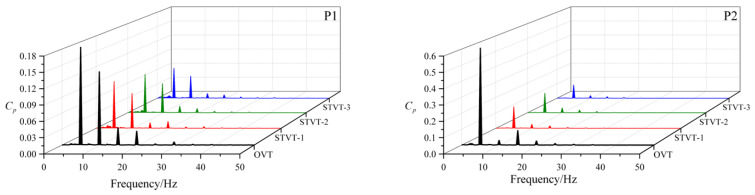
Frequency−domain pressure fluctuations at points P1 and P2.

**Figure 12 entropy-25-00545-f012:**
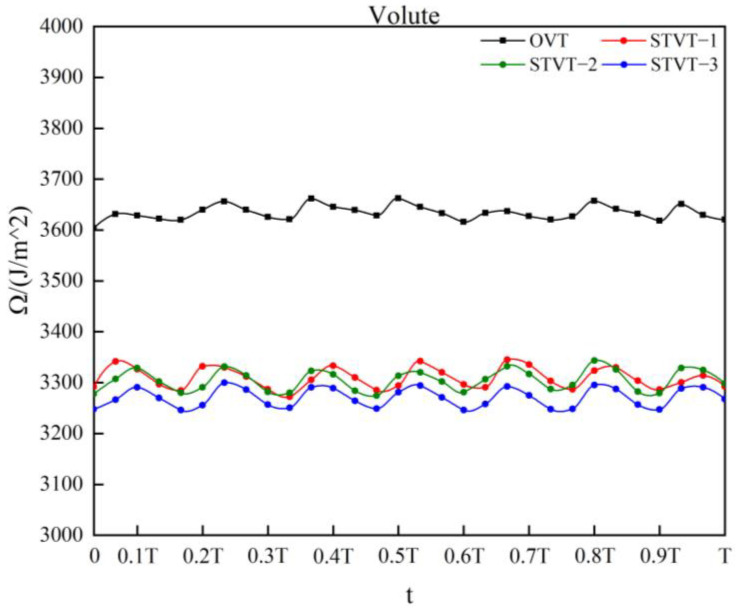
Total enstrophy of the flow field in volute of four pumps.

**Figure 13 entropy-25-00545-f013:**
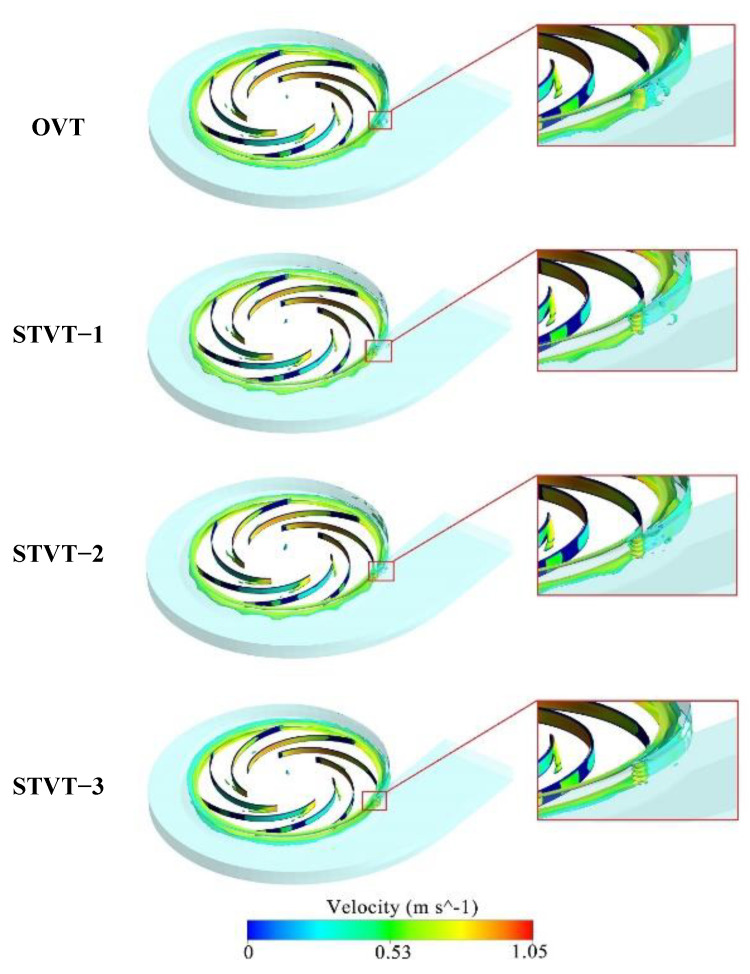
Total enstrophy per unit volume iso−surface of four model pumps.

**Figure 14 entropy-25-00545-f014:**
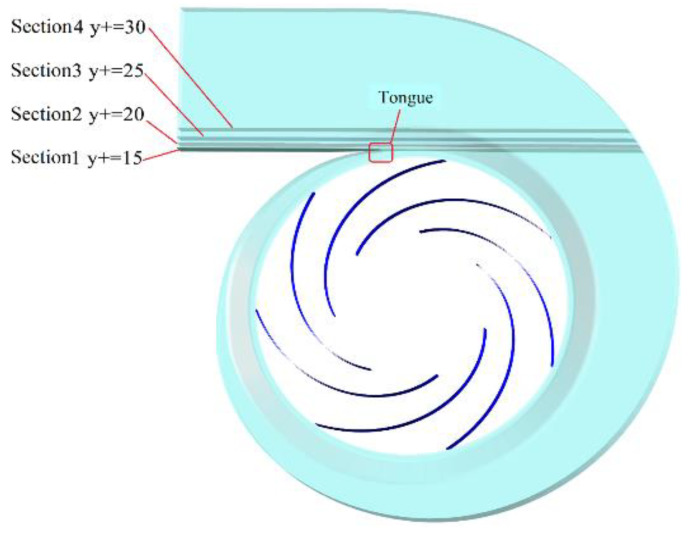
Different y^+^ section positions.

**Figure 15 entropy-25-00545-f015:**
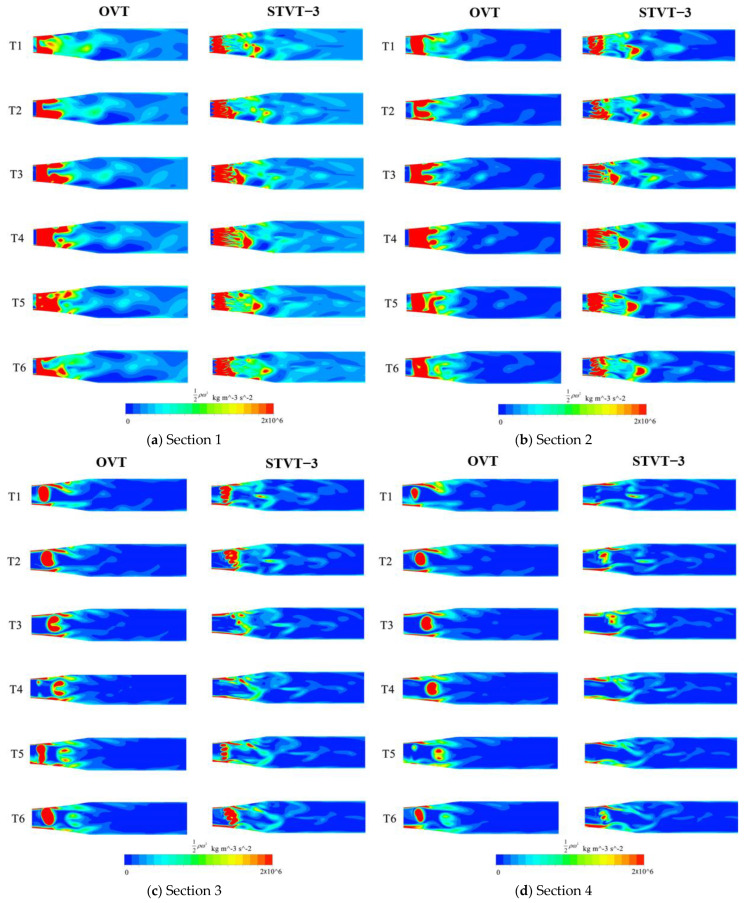
Enstrophy per unit volume distribution of model pumps OVT and STVT−3 in each section at T1−T6.

**Figure 16 entropy-25-00545-f016:**
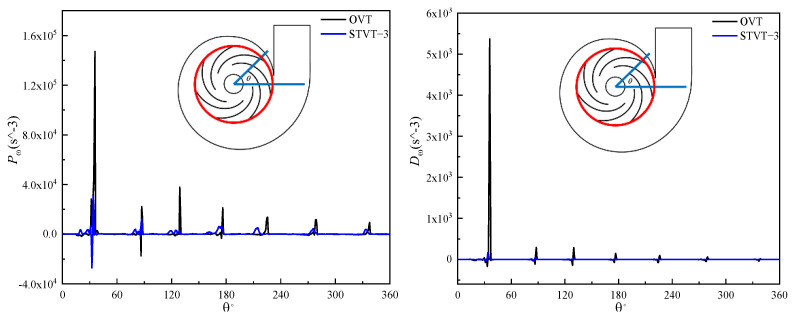
Curves of the enstrophy transport equation of the model pump at the impeller outlet.

**Figure 17 entropy-25-00545-f017:**
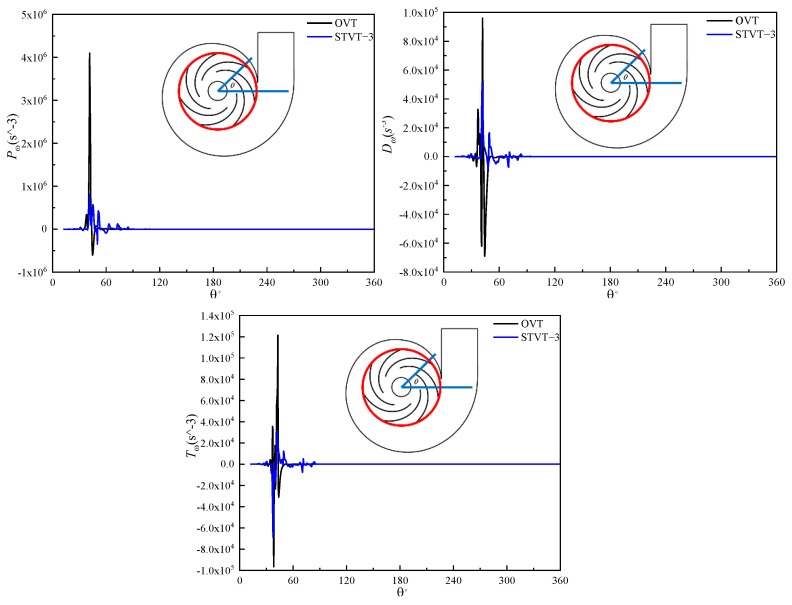
Curves of the enstrophy transport equation of the model pump at the volute wall.

**Figure 18 entropy-25-00545-f018:**
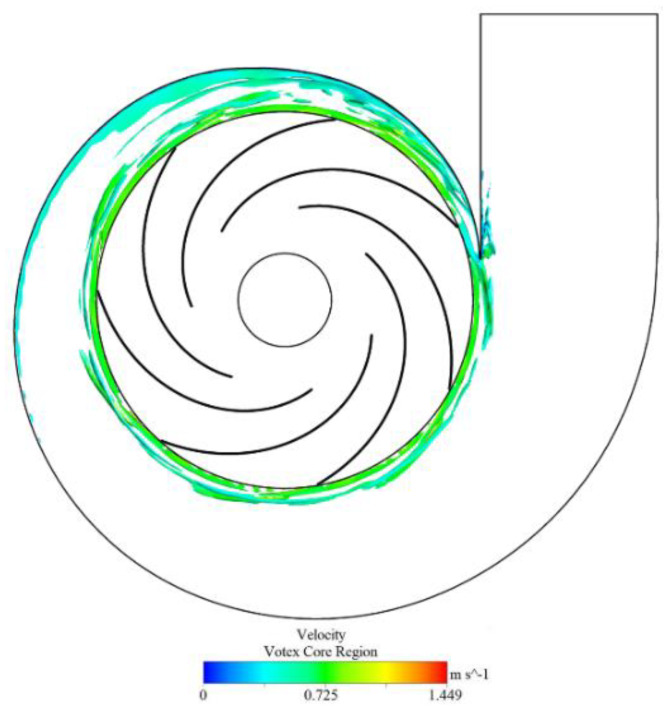
Iso−surface distribution of the production $P_{\omega }$ of the prototype pump OVT in the volute domain.

**Figure 19 entropy-25-00545-f019:**
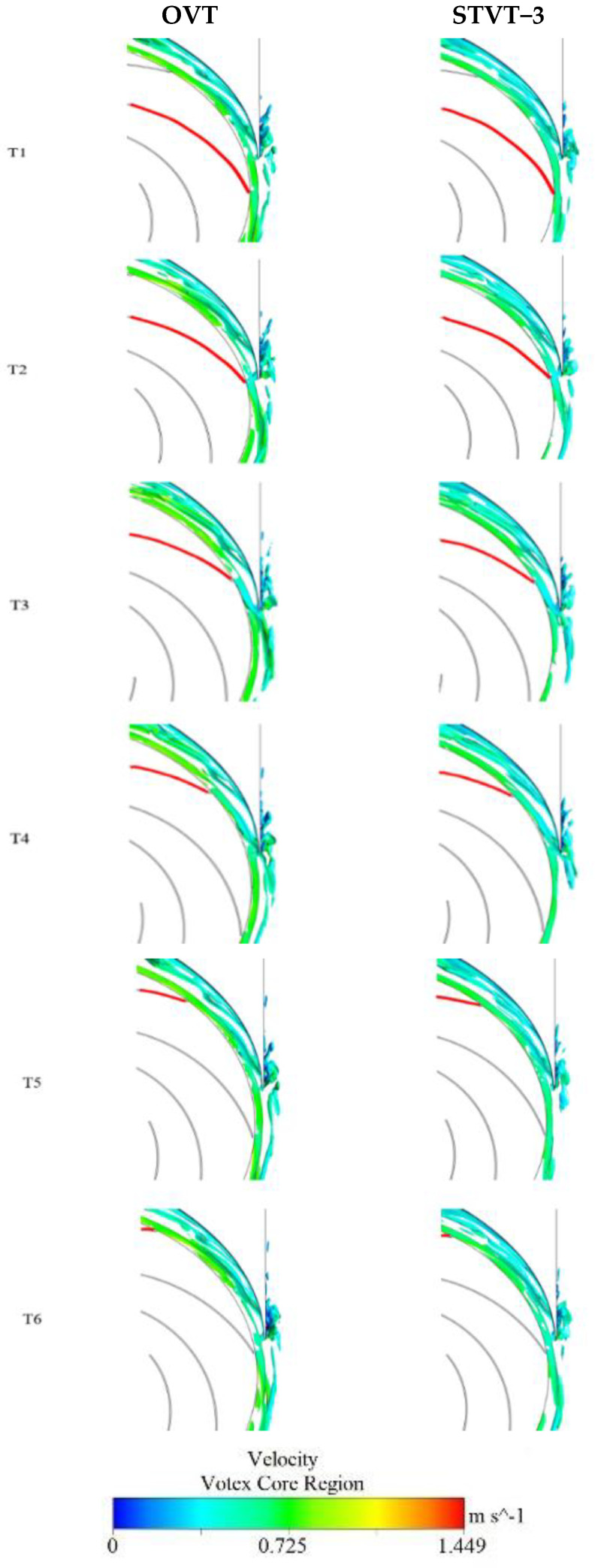
Distribution of production $P_{\omega }$ at different moments (T1–T6) in the tongue area.

**Table 1 entropy-25-00545-t001:** Parameters for the test centrifugal pump ([19]).

Parameters	Sign	Value
Nominal head coefficient	$\psi_{n}$	0.124
Inlet diameter of impeller (mm)	$d_{1}$	320
Outlet diameter of impeller (mm)	$d_{2}$	640
Blade width at exit (mm)	$b_{2}$	25
Blade angle at exit (°)	$\beta_{2}$	30
Blade number	Z	7
Volute tongue diameter (mm)	$d_{3}$	672
Inlet width of volute (mm)	$b_{3}$	25

**Table 2 entropy-25-00545-t002:** Parameters for the bionic tongues.

	The Number of Sinusoidal Tubercles	*λ*/(mm)	*A*/(mm)	*C*/(mm)	*L*/(mm)
STVT−1	4	7.5	1	25	15
STVT−2	4	7.5	2	25	15
STVT−3	4	7.5	3	25	15

**Table 3 entropy-25-00545-t003:** Numerical results for four different grid schemes.

Case	1	2	3	4
Nodes ($10^{6}$)	2.24	3.48	4.49	5.93
Mean $y^{+}$	16.3	10.5	7.2	5.8
Head coefficient $\psi^{*}$	0.1072	0.1065	0.1061	0.1057
Efficiency η	84.5	83.8	83.4	82.9

**Table 4 entropy-25-00545-t004:** Reduction of pressure fluctuation amplitude at BPF.

	Reduction (%)		
Monitoring Point	STVT−1	STVT−2	STVT−3
P1	42.34	45.67	51.28
P2	60.41	62.32	66.75

**Table 5 entropy-25-00545-t005:** Time−averaged enstrophy of four pumps and the reduction compared to the prototype pump.

	The Time−Averaged Total Enstrophy (J/m^2^)	Reduction (%)	Efficiency (%)
OVT	3633	0	83.4
STVT−1	3309	8.93	84.7
STVT−2	3304	9.07	85.1
STVT−3	3270	10.01	85.5

## Data Availability

Data sharing not applicable. No new data were created or analyzed in this study. Data sharing is not applicable to this article.

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
