# Peer review of "Analysis of the Energy Loss and Performance Characteristics in a Centrifugal Pump Based on Sinusoidal Tubercle Volute Tongue"

_entropy, 2023, doi:10.3390/e25030545_

Round 1
Reviewer 1 Report
please see comments attached

Author Response
On behalf of my co-authors, we thank you very much for giving us an opportunity to revise our manuscript, we appreciate two reviewers very much for their positive and constructive comments and suggestions on our manuscript entitled " Analyze of the energy loss and performance characteristics in a centrifugal pump based on sinusoidal tubercle volute tongue" (ID: entropy-2195929). We have studied two reviewers’ comments carefully and have tried our best to revise our manuscript according to the comments. The manuscript has been submitted to the editing system, which we would like to submit for your kind consideration. We would like to express our great appreciation to reviewer for comments on our paper.
Reviewer #1’s Comments:
The paper investigates the impact of volute tongue shape on a low specific speed pump performance. The authors use a numerical approach, using CFD and enstrophy to quantify the losses. The paper suffers from a number of shortcomings, however, and needs major revision for possible publication in the Journal. The authors should completely revise the paper taking account of the following comments:
1. The grammar and style need to be thoroughly improved. This concerns very much the introduction; but–at times–the entire text. Avoid commonplace statements: your readers are familiar with centrifugal pumps.
Response 1: Thank you for your comment. The article has been edited by a professional editing agency. The corresponding editing certificate has been uploaded through the system.
2. Make the text more concise: there are several repetitions where the same fact is repeated a 2nd or 3rd time after the 1st mention.
Response 2: Thank you for your comment. This issue has been revised in full context.
3. The quality of the figures should be improved to ease understanding.
Response 3: Thank you for your comment. This issue has been revised in full context.
4. Table 1: the volute tongue diameter is d3=672 mm (not 336 mm).
Response 4: Thank you for your comment. This is a mistake in writing.
5. The volute tongue leading edge is b3=30 mm (line 74, Fig. 2 dimension “C” and text). Table 1 gives b3=25 mm. In Fig. 1 it looks like the leading edge of the volute tongue is at r3=336 mm and the width would be b3=25 mm; but this seems not the case and contradicts the above. Rework Fig. 1 to clarify what is meant. b3 should correspond to the tongue leading edge, where also d3 is defined.
Response 5: Thank you for your comment. This is a mistake in writing.
6. Below profile shows the leading edge of the tongue as a cosine wave with amplitude A=3 mm and wave length of 7.5 mm. This should represent STVT‐3 (flow is from top): is that correct? 2A=6 mm.
Response 6: Thank you for your comment. The figure below shows that the leading edge of the tongue is a cosine wave is correct.
8. Lines 76, 77: the surface of the discharge channel is grooved over a length of 15 mm over which the depth of the grooves decreases from 2A to zero. For 2A=6 mm this seems quite deep; is that true? Only one side of the tongue grooved in this way? Longitudinal section of the tongue through the groove and through the “tubercle” where thickness is maximum?
Response 8: Thank you for your comment. We are basing this size of a sinusoidal volute tubercle based on a previous article, please see Reference 17.
- Lin, P. F.; Song, P. F.; Zhu, Z. C.; Li, X. J. Research on the Rotor-Stator Interaction of Centrifugal Pump based on Sinusoidal Tubercle Volute Tongue. Journal of Applied Fluid Mechanics. 2021, 14, 589-600.
9. Specify the specific speed and design head of the pump. With ψ*=0.124 and angular speed of 4.2 rad/s one gets a head of H=0.091 m, power P=10 W, torque M=2 Nm, u2=1.34 m/s, nq=22 (rough figures). Is this correct? The data look a little peculiar. The efficiency of 83% seems very high (expected 65 to 70%). Measurement accuracy might be a serious problem with these low values. This issue needs to be checked and addressed.
Response 9: Thank you for your comment. The geometry used in this study has been mentioned in several articles, and our simulation results are similar to those obtained by their simulations. Therefore, our data results have some credibility. Please see “Effect of the volute tongue profile on the performance of a low specific speed centrifugal pump”(DOI: 10.1177/0957650914562095), “Effects of Volute Curvature on Performance of a Low Specific-Speed Centrifugal Pump at Design and Off-Design Conditions”(DOI: 10.1115/1.4028766) and “Development of new ‘multivolute casing’ geometries for radial force reduction in centrifugal pumps”(DOI:
10.1080/19942060.2015.1004787).
10. Define flow coefficient or delete since not used in this paper.
Response 10: Thank you for your comment. This section has been corrected.
11. Define dimensionless velocities: V*=V/u2? If so, in Fig. 8 we would see maximum velocities reaching 1.4 u2 which is seems extremely high and unrealistic (the boundary layer thrown off an impeller could only reach u2 at maximum).
Response 11: Thank you for your comment. The velocity shown in Figure 8 is not dimensionless, but a post-processed velocity streamline plot with rulers corresponding to each other.
12. Flow traverses done in mid‐section of volute and impeller (i.e. at b2/2)?
Response 12: Thank you for your comment. This point is not mentioned in the article, and this article is a study of the degree of energy loss in the pump before and after modification. However, there is a tendency to flow traverse in the middle section of the volute and impeller
13. It is suggested to revise the structure of sections 3.3 and 4. Fig. 4 is found in line 120, but the pressure coefficient shown in Fig. 4 is only introduced in line 143. And Eq. 2. Pressure coefficient Δp is introduced in line 112 before fig. 4.
Response 13: Thank you for your comment. According to the comments, corrections have been made in the appropriate places. Please see section 3.3 in yellow.
14. The shape of the efficiency curves for all the 4 pumps is peculiar–due to the particular shapes of the impeller inlet and the volute. The reasons should be analyzed and explained from the CFD flow patterns.
Response 14: Thank you for your comment. The efficiency curve shapes of the four pumps in this paper are broadly consistent with the efficiency curve trends of previous literature that have been studied using this original model. Please see “Li, X., Gao, P., Zhu, Z. and Li, Y. Effect of the blade loading distribution on hydrodynamic performance of a centrifugal pump with cylindrical blades. Journal of Mechanical Science and Technology, 2018. 32(3), 1161–1170. DOI: 10.1007/s1 2206-018-0219-4.”
15. Fig. 7: Provide measured efficiency and power curves compared to calculated curves. In Fig. 7, at Q/Qd=1.3, STVT‐2 has the highest head and efficiency. This is does not agree with lines 148 to 153. Make sure that the colors and symbols of the curves used in the graphs can be clearly recognized (sometimes graphs copied from MS‐Excel change colors).
Response 15: Thank you for your comment. This is a post-processing error. Corrections have been made in the appropriate places according to the comments. Please see Figure 7.
16. Line 165: …at the mid cross section of the pump?
Response 16: Thank you for your comment. Yes, our pressure contour and velocity streamline are generated based on the mid-section of the pump.
17. Fig. 8: what is shown: cp or V? You cannot show both in the same picture by color grading.
Response 17: Thank you for your comment. The upper left corner of Figure 8 shows the pressure coefficients, while the lower right corner shows the velocity streamline, and the rulers of these two figures are represented separately. We've added a clear note in the original article.
18. Fig. 9: monitor point P2 is right on the leading edge of the tongue; please confirm and mention in the text for clarity and emphasis. Is monitor point P1 inside the discharge channel or inside the volute facing the impeller. This is not clear from the picture.
Response 18: Thank you for your comment. P1 was set inside the volute facing the impeller and P2 was set right on the leading edge of the tongue. We've added the appropriate instructions as required. Please see section 4.1.3 in yellow.
19. The pressure fluctuations in the time domain in Fig. 11 are difficult to read and understand because the curves cannot be distinguished easily.
Response 19: Thank you for your comment. It may be due to a typographical error that causes Figure 11 to coincide with Figure 10. Corrections have been made, as shown in Figure 11.
20. Why not show measured pressure pulsations?
Response 20: Thank you for your comment. It may be due to a typographical error that causes Figure 10 to coincide with Figure 11. Corrections have been made, as shown in Figure 10.
21. Table 5: give dimension of enstrophy; add a column with the measured efficiency.
Response 21: Thank you for your comment. According to the comments, corrections have been made in the appropriate places. Please see Table 5 in yellow.
22. Wall friction also increases energy dissipation. Is this contribution missed when integrating vorticity?
Response 22: Thank you for your comment. According to the actual application of the engineering, wall friction will indeed increase energy loss. However, in this study, the influence of wall friction was ignored in the numerical simulation. The influence of wall friction on energy loss will be considered in subsequent studies.
23. Can you provide a relation between enstrophy and losses? Could the local power loss be referred to the hydraulic power delivered by the pump?
Response 23: Thank you for your comment. The relationship between enstrophy and loss is indeed a key element in the study of energy loss within the pump. And with the weakening of energy loss, the local power does increase to a certain extent. As you can see, the original model compared in this article is a reference due to insufficient laboratory resources. But please give me a valuable opportunity to continue my research based on this research. It is currently on the agenda.
- Would y+=15 be in the order of y=0.3mm? If this is the order of magnitude, I could not so well imagine “strong turbulent fluctuations” as you write in line 279. Intuitively, I rather would think of “shear” in so thin a fluid layer near a solid wall.
Response 24: Thank you for your comment. This part is a writing error. This section has been removed.
- Fig. 13: the figure capture refers to enstrophy while the color scale shows velocity. Similar in Fig. 18; please ensure consistency.
Response 25: Thank you for your comment. Figure 13 shows the velocity isosurface distribution of the total quasi-vortex energy per unit volume of the full-flow field and tongue septum region of the six model pumps.
- Line 309: what do you understand by “expansion and viscosity fluctuation”? Expansion of what? Eddy viscosity (since the viscosity of the water is constant)?
Response 26: Thank you for your comment. The main meaning expressed here is to avoid the impact of the occurrence of shear thickening effect on this study. The shear thickening effect is that the viscosity value of the fluid increases when the shear rate increases, and the viscosity value decreases when the shear rate decreases
- Lines 318, 319: Molecular diffusion reduces enstrophy, but increases losses (temperature rises imperceptibly).
Response 27: Thank you for your comment. is the molecular dissipation term in the enstrophy transport equation, which is always negative, so this term has a decreasing effect on the enstrophy.
- If I understand correctly, Figs. 16 and 17 present a momentary picture (like a photograph). That is why we see only the one peak of the blade which is closed to the tongue. In contrast, Figs. 11 and 12 provide records over one or two revolutions.
Response 28: Thank you for your comment. Yes, your understanding is exactly correct. I feel very honored.
- The topic of the paper “sinusoidal leading edge (LE) of the volute tongue” is quite interesting. Oval shapes of the tongue leading edge have been often applied (as well as oblique LE) to reduce RSI. Did you compare your data with oval LE? The oval LE would be more easily to manufacture. The reduction of RSI brought about by oval or oblique tongue leading edge is because the wakes flowing off the blades are “smeared” over some circumferential distance. This same effect operates also with the cosine‐shaped LE. Several “tubercles” may not be needed and the effect may be quite different from those attributed to the humpback whale.
Response 29: Thank you for your comment. We have also found some leading edges of oval diaphragms used in centrifugal pumps, and indeed as you said, this will be easier to manufacture. Again, as you said, it's really significant to compare that, and that's what we're doing. At present, we have such a National Natural Science Foundation project, the content is to make sinusoidal shaped diaphragm more advantageous in the practical engineering application of centrifugal pumps, and easy to put into large-scale production. Of course, this part will be presented in more detail in the near future.
- The volute tongue of the test pump seems quite long and thin. The leading edge may have semi‐circular shape. Such a design is very sensitive to incidence and prone to generate high pressure fluctuations. The shape of the tongue should be clearly reported because it is vital for correctly appreciating the results of the study. It is suspected that the volute tongue design might have contributed strongly to the unusual shape of the efficiency curves.
Response 30: Thank you for your comment. The test pump used in this study is a reference for introduction, and the significance of this study is to explore whether this bionic modification has a role in improving the amplitude of pressure pulsation and energy loss. My co-authors and I very much hope that you will give us a chance to further refine the model soon, so stay tuned.

Reviewer 2 Report
There are some suggestions as below.
Q1. The main geometric parameters of the experimental pump in Table 1 are inconsistent with those in Ref. [18].
Q2. The parameters of the bionic tongue in Table 2 are not available in units.
Q3. For the convenience of non-specialist readers, numerical model including governing equations, numerical method and boundary conditions should be given in your paper.
Q4. In the time step setting, how many seconds is one time step?
Q5. The DES model has an inherent defect, grey area problem. Why the authors didn’t use the enhanced DES-type model such as DDES and IDDES? Please comment.
Q6. Please consider to revise the sentence " As shown in Table 3, case 3 can best predict the performances that agree with the references..." in page 4.
Q7. Please explain the reason for the deviation between the experimental data and the calculated results.
Q8. The second sentence of the first paragraph on page 7, " The relative increase was 7% (STVT-1), 8.1% (STVT-2), and 9.2% (STVT-3) at 1.32Qd. " This sentence does not match the figure.
Q9. Page 7. In my opinion, the region with high pressure pulsation cannot be determined by the distribution of pressure coefficient.
Q10. Some figures are not clear. For example, the legend in fig.15.
Q11. The English in the manuscript needs improvement in terms of grammar or spelling
Q12. In the results section, an in-depth discussion should be made for enhancing the practical implication of the results and findings from current study, which will be more useful than that for only performing the statistical data collection and analysis.
Author Response
On behalf of my co-authors, we thank you very much for giving us an opportunity to revise our manuscript, we appreciate two reviewers very much for their positive and constructive comments and suggestions on our manuscript entitled " Analyze of the energy loss and performance characteristics in a centrifugal pump based on sinusoidal tubercle volute tongue" (ID: entropy-2195929). We have studied two reviewers’ comments carefully and have tried our best to revise our manuscript according to the comments. The manuscript has been submitted to the editing system, which we would like to submit for your kind consideration. We would like to express our great appreciation to reviewer for comments on our paper.
Reviewer #2’s Comments:
Q1. The main geometric parameters of the experimental pump in Table 1 are inconsistent with those in Ref. [18].
Response 1: Thank you for your comment. This is a writing error. Corrected in the appropriate place.
Q2. The parameters of the bionic tongue in Table 2 are not available in units.
Response 2: Thank you for your comment. We're sorry this was an oversight at the time of writing, and we've added it where appropriate.
Q3. For the convenience of non-specialist readers, numerical model including governing equations, numerical method and boundary conditions should be given in your paper.
Response 3: Thank you for your comment.
Q4. In the time step setting, how many seconds is one time step?
Response 4: Thank you for your comment. In this study, a time step is the time it takes for the impeller to rotate by one degree, and the time value is 0.0041556 second.
Q5. The DES model has an inherent defect, grey area problem. Why the authors didn’t use the enhanced DES-type model such as DDES and IDDES? Please comment.
Response 5: Thank you for the valuable comment. Indeed, as you said, the current DES model is flawed. However, the main research content of this paper is the effect of sinusoidal tubercles on pressure pulsation and energy loss in centrifugal pump, and the DES model is relatively sufficient, and it can also save some time.
Q6. Please consider to revise the sentence " As shown in Table 3, case 3 can best predict the performances that agree with the references..." in page 4.
Response 6: Thank you for your comment. Corrected in the appropriate place in yellow.
Q7. Please explain the reason for the deviation between the experimental data and the calculated results.
Response 7: Thank you for your comment. Ignoring the influence of wall friction and external temperature changes in the simulation is the root cause of the error between the simulation and the experiment.
Q8. The second sentence of the first paragraph on page 7, " The relative increase was 7% (STVT-1), 8.1% (STVT-2), and 9.2% (STVT-3) at 1.32Qd. " This sentence does not match the figure.
Response 8: Thank you for your comment. This is a post-processing error. Figure 7 has been modified.
Q9. Page 7. In my opinion, the region with high pressure pulsation cannot be determined by the distribution of pressure coefficient.
Response 9: Thank you for your comment. The high-pressure pulsations that appear in the seventh page statement are described as a typographical error.
Q10. Some figures are not clear. For example, the legend in fig.15.
Response 10: Thank you for your comment. The purpose of this section is to show the reader how the sinusoidal tubercle volute tongue improves the flow field in the pump.
Q11. The English in the manuscript needs improvement in terms of grammar or spelling.
Response 11: Thank you for your comment. My English scientific paper writing level really needs to be improved, and I am actively working to improve it. Moreover, our manuscripts have been polished in well-known institutions. The certificate of polishing of this manuscript is shown below.
Q12. In the results section, an in-depth discussion should be made for enhancing the practical implication of the results and findings from current study, which will be more useful than that for only performing the statistical data collection and analysis.
Response 12: Thank you for your comment. We have made corresponding changes in the conclusion section.

Round 2
Reviewer 1 Report
Authors response to first review:
1. Errors concerning actual figures and values have been corrected. That's okay.
2. The more technical issues have not been addressed properly because the results seem not be well understood.
3. Referring the questions to earlier publications on the same pump is not of much help. I'm not willing to study 3 more papers. Nor do I want to by a paper for the review which is not open source.
4. Referring to future studies may do when real scientific issues are at stake but not for issues which could have easily resolved in the frame of the present paper.
5. The same pump has been studied in former publications. There are repetitions which could be avoided.
Reviewer 2 Report
The paper has been carefully revised and polished by the authors. The quality of the article has been improved and many deficiencies in the original manuscript have been made up. In a whole, the paper can be published in current version,